# Investigation of Mental and Physical Health of Nurses Associated with Errors in Clinical Practice

**DOI:** 10.3390/healthcare10091803

**Published:** 2022-09-19

**Authors:** Despoina Pappa, Ioannis Koutelekos, Eleni Evangelou, Evangelos Dousis, Georgia Gerogianni, Evdokia Misouridou, Afroditi Zartaloudi, Nikoletta Margari, Georgia Toulia, Polyxeni Mangoulia, Eftychia Ferentinou, Anna Giga, Chrysoula Dafogianni

**Affiliations:** 1Department of Nursing, University of West Attica, 12243 Athens, Greece; 2Department of Nursing, Evangelismos General Hospital, 10676 Athens, Greece

**Keywords:** mental health, physical health, resilience, nurses, errors

## Abstract

Background: Errors are common among all healthcare settings. The safety of patients is linked directly with nursing errors because nurses stand by them more often than any other healthcare professional. The role of mental and physical health of nurses is of great interest for a good and efficient job performance, but also for maintaining good patient care delivery. This study aimed to investigate the association between nurses’ general health and making errors during clinical practice. Methods: A total of 364 nurses completed a specially designed questionnaire anonymously and voluntarily. The sample consisted of nurses with all educational degrees. The questionnaire included demographic data and questions about general health issues, resilience status and nurses’ possible experience with errors within a hospital. Results: 65,8% of the participants stated that at least one error had happened at their workplace, and 49,4% of them reported that the error was caused by them. Somatic symptoms were found to have a positive correlation with making errors (*p* < 0.001). However, the other aspects of general health, which were anxiety/insomnia, social dysfunction and severe depression, had no statistical significance with adverse events. The most common type of error reported (65,5%) was a medication adverse event. Resilience level was found to be statistically significant (*p* < 0.001) when correlated with all aspects of general health (anxiety/insomnia, severe depression, somatic symptoms), but not with social dysfunction. Conclusion: Nurses are affected by their somatic symptoms in their daily clinical practice, making them vulnerable to making errors that compromise patient safety. A high resilience level could help them cope with unfavorable situations and prevent them from doing harm to a patient or themselves.

## 1. Introduction

The incidence of errors is high worldwide [1,2,3], with reports indicating that 1 out of 10 patients is affected during their hospitalization [4,5,6]. Furthermore, for a percentage of approximately 7%^5^, these mistakes have irreversible consequences. The World Alliance for Patient Safety adds that 10% of hospitalized patients in developed countries experience an adverse event annually. At the same time, there is increasing concern about reversible deaths because of in-hospital errors [7].

A nursing error involves an unintended “accident” made by a nurse that adversely affects—or could adversely affected—the safety and quality of care of a patient [8]. According to the Nurses Ethical Codes [9], nurses have an important role in safeguarding the integrity of patients, as they spend more time with them than any other healthcare professional. Therefore, most errors are more likely to be made by the nursing staff within a hospital. In a large research study where more than 43,000 nurses participated [10,11], the essential magnitude of the problem was shown, as the factors of burnout, understaffing, non-observance of duties and insufficient nursing support contributed dramatically to the improper delivery of health care. The most common errors within a hospital are infections, falls, pressure ulcers, medication errors, documentation errors and equipment injuries [12].

Nurses are particularly affected by the workplace. Stress and constant association with patients who are dying or suffering directly affect their mental health. Shift work that affects the circadian rhythm [13] as well as workload have been extensively studied in terms of their impact on the physical condition of nurses [14,15,16,17,18,19,20,21]. The hospital’s largest workforce, which works under a shift system, has been shown to have a higher rate of cardiovascular diseases, diabetes, dementia, sleep and weight disorders, obesity and more [13,22]. Olds& Clarke [23] reported that for every hour of work, the possibility of the wrong drug or the wrong dose increases by 2%. Along with prolonged burnout and a less than optimal state of well-being in nurses, the quality of care provided is inadvertently affected [24,25]. The management of psychological and physical stress, as well as the adoption of good health practices, are issues of major importance for nurses in terms of the effective performance of their duties, since they are the ones that directly affect the quality-of-care provision [14,21]. This study aimed to investigate the association between nurses’ general health and making errors during clinical practice.

## 2. Materials and Methods

### 2.1. Participants

The study participants consisted of nurses with all educational degrees, such as university nurses, nurses from technological institutions and assistant nurses of secondary education. The population of the present study was working at general hospitals.

### 2.2. Data Collection

The present study was a cross-sectional study performed through completion of an anonymous and voluntary questionnaire from November 2019 to November 2020. The research was approved by the Ethics Committee of the University of West Attica and the scientific councils of all the hospitals involved. Due to restrictive measures for the worldwide pandemic from hospitals’ policies within the total study period, it was necessary to create a different way to distribute and collect questionnaires, so an electronic form of the tool was developed too.

### 2.3. Instruments

The research tool consisted of four sections: 1. The demographic data: questions concerning the demographic and occupational status of the participants, such as gender, age, marital and educational status, working section (inpatient nurse/outpatient nurse/operating nurse/oncology nurse/other) and duration of work under a specific unit, 2. The General Health Questionnaire (GHQ-28) [26], which describes a 4-factor structure (somatic symptoms, anxiety/insomnia, social dysfunction and depression), 3. The Taxonomy of Error, Root Cause Analysis and Practice-responsibility (TERCAP) [27], which is designed to collect nursing practice breakdown data from boards of nursing. It describes a set of categories that is based on notions of good nursing practice, such as Safe Medication Administration, Documentation, Surveillance, Prevention, Intervention, Clinical Reasoning, Interpretation of Orders and Professional Responsibility, and 4. The Brief Resilience Scale (BRS) [28], which is a 5-point Likert scale about six specific statements of daily life routine.

### 2.4. Data Analysis

In this study, quantitative variables were initially tested for normality using the Kolmogorov–Smirnov criterion. The same variables were expressed as mean (SD = Standard Deviation) or median (interquantile range), absolute and relative frequencies. Student’s *t*-tests were computed for the comparison of mean values. Multiple linear regression analysis was used with the dependentGHQ-28 scores. The regression equation included terms for participants’ demographics, work-related characteristics and the occurrence of an error during their work. Adjusted regression coefficients (β) with standard errors (SE) were computed from the results of the linear regression analyses. In this study, the *p*-values were two-tailed. Statistical significance was set at *p* < 0.05. The analysis was accomplished using SPSS-version 22.0 statistical software.

## 3. Results

### Demographic Characteristics

In this study, 364 nurses were included. Their demographics and occupational characteristics are presented in Table 1. Most of the participants were women (87.3%), were aged from 22 to 35 years (43.6%), were married or living with their partner (50%) and had children (45.6%). Moreover, 48% of the sample had a university degree, 10.2% were specialized, 50.5% had a monthly income of EUR 500–1000, 12.1% had a second job and almost all (94.7%) were Greek native speakers. The median number of years of working experience in their present hospital was 9 years (IQR: 1–15 years). The mean resilience score was 20.4 (SD = 4.2).

Almost 7 out of 10 participants (65.8%) had experienced an error in their job. More specifically, 49.4% of the participants stated that they had caused an adverse event themselves, and 73.2% that someone else had caused it. The most frequent places that errors occurred were the room of the patient (29.2%) and the Intensive Care Unit (ICU) (24.3%). Most of the errors were made on male patients (65.1%), and in 32.1% of the cases, there was a complication in the patient’s health after the error. The error involved some kind of intention or criminal behavior in only 6.7% of the cases, and the error involved a drug error in 65.5% of the cases.

Participants’ scores on the GHQ-28 subscales, as well as their total scores, are presented in Table 2. Somatic symptoms’ scores were found to differ significantly between participants who had experienced an error in their work and those who had not (*p* = 0.030). More specifically, participants who had experienced an error in their work had significantly greater somatic symptoms. All the other GHQ-28 subscales, as well as total score, were similar in both participants’ groups.

The difference on the somatic symptoms scale between participants who had experienced an error in their work and those who had not remained significant after adjusting for all other demographical and occupational characteristics (Table 3).

In addition, participants who were 36–45 years old had fewer somatic symptoms compared (*p* = 0.050) to those who were 22–35 years old, and participants who had greater resilience also had fewer somatic symptoms. Furthermore, participants with children had more anxiety/insomnia symptoms. On the other hand, participants who were 36–45 years old (*p* < 0.001) or more than 46 years old (*p* = 0.003) had fewer anxiety/insomnia symptoms compared to participants who were 22–35 years old. Participants who had greater resilience were found to experience fewer anxiety/insomnia symptoms.

Social dysfunction was not associated with any of the demographical and occupational characteristics (Table 4). On the contrary, participants with greater resilience were significantly associated with less severe depression symptoms (*p* < 0.001).

Participants who were 36–45 years old (*p* = 0.003) or more than 46 years old had better total GHQ-28 total scores compared to participants who were 22–35 years old (*p* = 0.013) (Table 5). Greater resilience of participants was significantly associated with better total GHQ scores (*p* < 0.001).

## 4. Discussion

This study aimed to examine the relationship between nurses’ physical and mental health status, resilience and occurrence of errors during daily job practice. A significant correlation was found between physical symptoms and occurrence of adverse events for nurses who participated in the research. Similar results were found by Arimura et al. [29], who indicated that there was a statistical significance between poor physical health and errors.

This study also showed that one third of the nurses reported that they were making errors mostly on day or evening shifts. Similarly, Gold et al. [30] found in their study that nurses working on rotating shifts and occasionally at night mentioned more medication errors due to insufficient sleep management.

Additionally, in the present study, one third of the participants referred to errors made while changing shifts (patient hand-offs). These results are congruent with those of Drach-Zahavy and Hadid [31], who examined handover communication between nurses and the types of errors happening at shift change. More specifically, inaccurate drug dosage and missing documentation were found to be the top errors reported at that time.

The present study also found that errors were significantly correlated with physical fatigue and anxiety. Similarly, West et al. [32], who investigated in-hospital errors associated with fatigue, anxiety and insomnia, found relevant findings, since nurses’ fatigue was provoked by heavy workload. When the workload was consistent, fatigue and burnout were regarded as chronic. In this study, also, eight out of ten nurses stated that the workload of nursing staff negatively affected the occurrence of their reported error. According to Al-Kandari and Thomas [33], adverse events had a significant correlation with workload, resulting mostly in medication delays or omissions.

The findings of this study indicate that patients’ wards and ICUs are the most risky departments for an adverse event to happen. It is of great importance to mention that several studies [34,35] have investigated the occurrence rate of nursing errors within hospital units. In ICUs, the seriousness of patients’ medical conditions makes nursing practice more specialized, focused, demanding and exhausting. A recent study by Melnyk et al. [36] examined the association between errors and nurses’ mental and physical health. Their results indicated that when nurses were in poor mental and physical health, reporting an error was much more statistically significant. However, according to “Project to collect medical near miss/adverse events information”, [37] nurses make errors 0.56 times more often in outpatient departments. Similarly, in Melnyk’s study [36], it was found that nurses in poor physical and mental health presented with 26% higher likelihood of making errors and 71% higher possibility of having an adverse event. They also stated that depression had a significant association with errors.

The present study also showed that the association of nurses’ general health with level of resilience had a positive impact on somatic symptoms, anxiety/insomnia and severe depression. So, when nurses were more resilient, they presented fewer physical symptoms. This finding was congruent with those of Koen et al. [38], who examined the prevalence of resilience in professional nurses, stating that low levels of mental discomfort were presented by resilient nurses. However, Koen et al. [38] indicated that although half of the nurses were flourishing with regards to their general health, the other half were not. So, they might need special support for social well-being and job satisfaction. It is important to recognize the actual needs of health professionals οn time.

## 5. Limitations

A part of this study was conducted during the first wave of the COVID-19 pandemic. Due to the restrictive measures on hospital access, electronic questionnaires were distributed to nursing staff after their agreement. This situation had the impact of eliminating direct communication with the personnel and further explanation of the study purpose and necessity. There was phone contact with the head nurses to obtain information about the proper execution of the procedure.

## 6. Conclusions

Nurses were affected mostly by their physical health in making errors in their daily practice. Fatigue, headaches, sickness tendency and exhaustion were the main descriptions for poor physical status in nurses in this study. A significant association was reported between nursing errors and somatic symptoms. Although several researchers associated errors with poor mental health, this study revealed no such connection. However, a prolonged stay of physical symptoms could lead to disrupted mental status, correlating errors with mental health in that way. Resilience might be a useful capacity to be obtained and developed in the nursing population, as it was shown that the more resilient a nurse was, the less somatic symptoms they had. This study would be necessary to continue due to the increased need for errors examination.

## Figures and Tables

**Table 1 healthcare-10-01803-t001:** Participants’ characteristics.

	N (%)
Gender	
Men	46 (12.7)
Women	318 (87.3)
Age	
22–35	159 (43.6)
36–45	132 (36.2)
46+	73 (20.2)
Married/Living with partner	182 (50)
Children	166 (45.6)
Educational level	
High school/secondary education	36 (9.9)
Two-year college graduate	27 (7.4)
University alumni	175 (48)
MSc/PhD holder	126 (34.7)
Specialized	37 (10.2)
Monthly income	
EUR 500–EUR 1000	184 (50.5)
EUR 1001– EUR 1500	170 (46.7)
EUR 1501– EUR 2000	9 (2.5)
EUR 2001 and above	1 (0.3)
Second job	44 (12.1)
Greek native speaker	345 (94.7)
Years of experience in present hospital, median (IQR)	9 (1–15)
Total number of beds in use in your unit, median (IQR)	12 (7–20.5)
Total number of beds in your unit, median (IQR)	14.5 (9–30)
Brief Resilience Score, mean (SD)	20.4 (4.2)

**Table 2 healthcare-10-01803-t002:** GHQ-28 subscales by total sample, and by having an error occur in the workplace.

	Total Sample		During Your Professional Career, Has Any Error Ever Occurred in Your Workplace?				
			No (N = 124; 34.1%)		Yes (N = 240; 65.9%)		
	Mean	SD	Mean	SD	Mean	SD	*p* Student’s *t*-Test
Somatic symptoms	7.67	4.64	6.93	4.48	8.06	4.68	**0.030**
Anxiety/insomnia	7.48	5.15	6.93	5.30	7.77	5.07	0.147
Social dysfunction	8.31	3.66	8.11	4.13	8.42	3.39	0.435
Severe depression	2.23	3.54	2.10	3.69	2.30	3.46	0.617
Total GHQ-28 score	25.78	12.64	24.18	11.53	26.58	13.11	0.094

**Table 3 healthcare-10-01803-t003:** Multiple linear regression results with somatic symptoms and anxiety/insomnia subscales as dependent variables.

	Somatic Symptoms			Anxiety/Insomnia		
	β ^+^	SE ^++^	*p*	β ^+^	SE ^++^	*p*
Gender						
Men						
Women	1.22	0.80	0.130	0.34	0.80	0.670
Age						
22–35						
36–45	−1.39	0.71	**0.050**	−2.56	0.73	**<0.001**
46+	−0.99	1.12	0.375	−3.46	1.15	**0.003**
Married/Living with partner						
No						
Yes	0.44	0.69	0.525	−0.28	0.70	0.688
Children						
No						
Yes	0.56	0.76	0.456	1.78	0.78	**0.022**
Educational level						
High school graduate/Two-year college graduate						
University alumni	−0.95	0.78	0.226	−0.65	0.81	0.423
MSc/PhD holder	0.36	0.82	0.664	1.13	0.85	0.185
Specialized						
No						
Yes	0.50	0.93	0.595	−0.36	0.96	0.709
Monthly income						
EUR 500-EUR 1000						
EUR 1001 and above	−0.13	0.60	0.824	−0.38	0.61	0.540
Second job						
No						
Yes	1.43	0.75	0.057	1.21	0.77	0.118
Greek native speaker						
No						
Yes	1.02	1.59	0.522	0.56	1.64	0.732
Years of experience in present hospital	−0.01	0.05	0.879	0.05	0.05	0.256
Total number of beds in use in your unit	−0.003	0.019	0.873	−0.001	0.020	0.953
Total number of beds in your unit	0.001	0.014	0.935	0.002	0.014	0.893
Brief Resilience Score	−0.39	0.06	**<0.001**	−0.61	0.06	**<0.001**
During your professional career, has any error ever occurred in your workplace?						
No						
Yes	1.05	0.54	**0.050**	0.42	0.55	0.449

^+^ Regression coefficient; ^++^ Standard Error.

**Table 4 healthcare-10-01803-t004:** Multiple linear regression results with social dysfunction and severe depression subscales as dependent variables.

	Social Dysfunction			Severe Depression		
	β ^+^	SE ^++^	*p*	β ^+^	SE ^++^	*p*
Gender						
Men						
Women	0.32	0.65	0.624	0.56	0.60	0.349
Age						
22–35						
36–45	−1.04	0.59	0.080	−0.53	0.54	0.325
46+	−1.11	0.94	0.236	−1.66	0.86	0.054
Married/Living with partner						
No						
Yes	0.70	0.57	0.225	0.07	0.53	0.889
Children						
No						
Yes	−0.42	0.63	0.508	0.45	0.58	0.439
Educational level						
High school graduate/Two-year college graduate						
University alumni	−0.89	0.65	0.173	−0.68	0.60	0.257
MSc/PhD holder	−0.94	0.69	0.170	0.13	0.63	0.842
Specialized						
No						
Yes	0.54	0.78	0.487	−0.07	0.72	0.922
Monthly income						
EUR 500–EUR 1000						
EUR 1001 and above	−0.70	0.50	0.164	−0.72	0.46	0.119
Second job						
No						
Yes	−0.48	0.63	0.444	0.09	0.58	0.879
Greek native speaker						
No						
Yes	−1.65	1.34	0.217	0.48	1.22	0.698
Years of experience in present hospital	0.03	0.04	0.498	0.05	0.04	0.130
Total number of beds in use in your unit	0.008	0.016	0.606	−0.019	0.015	0.197
Total number of beds in your unit	0.005	0.012	0.641	0.013	0.011	0.223
Brief Resilience Score	−0.06	0.05	0.255	−0.25	0.05	**<0.001**
During your professional career, has any error ever occurred in your workplace?						
No						
Yes	0.20	0.45	0.657	0.42	0.41	0.305

^+^ Regression coefficient; ^++^ Standard Error.

**Table 5 healthcare-10-01803-t005:** Multiple linear regression results with total GHQ-28 score as a dependent variable.

	Total Score GHQ-28		
	β ^+^	SE ^++^	*p*
Gender			
Men			
Women	3.49	2.09	0.095
Age			
22–35			
36–45	−5.49	1.85	**0.003**
46+	−7.24	2.91	**0.013**
Married/Living with partner			
No			
Yes	1.11	1.79	0.535
Children			
No			
Yes	2.31	1.97	0.241
Educational level			
High school graduate/Two-year college graduate			
University alumni	−3.56	2.06	0.085
MSc/PhD holder	0.35	2.15	0.873
Specialized			
No			
Yes	0.56	2.43	0.818
Monthly income			
EUR 500–EUR 1000			
EUR 1001 and above	−1.77	1.56	0.256
Second job			
No			
Yes	2.42	1.96	0.216
Greek native speaker			
No			
Yes	0.35	4.15	0.934
Years of experience in present hospital	0.13	0.12	0.293
Total number of beds in use of your unit	−0.014	0.050	0.779
Total number of beds of your unit	0.022	0.036	0.546
Brief Resilience Score	−1.28	0.16	**<0.001**
During your professional career, has any error ever occurred in your workplace?			
No			
Yes	2.34	1.40	0.097

^+^ Regression coefficient; ^++^ Standard Error.

## Data Availability

All the data generated during this study are included in this published article.

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
