# Peer review of "Investigation of Mental and Physical Health of Nurses Associated with Errors in Clinical Practice"

_healthcare, 2022, doi:10.3390/healthcare10091803_

Round 1

Reviewer 1 Report

Dear Author

I have reviewed your manuscript "Investigation of mental and physical health of nurses associated to errors in clinical practice". I need some clarifications.

*The study participants consisted of nurses of all educational degrees such as university nurses, nurses form technological institutions and assistant nurses of secondary education. Does this affect your study results?

*In the tools, you used The Taxonomy of Error, Root 86 Cause Analysis and Practice-responsibility (TERCAP). But you did not explain about this in your results. But I found some explanation in discussion. Since you have described this tool in methods, provide the results as well. 

*Line 106 - you mentioned as 364 health professionals. Health professionals include physicians, pharmacists, nurses, etc. Since you have included only nurses, specify as 364 nurses.

*Table 1 - results does not match with the sample size. Gender - total was only 361; Age - 362; Educational level - 400. In table 2, the total was 363. I am afraid this change would affect other tables and your study results. Recheck.

Detailed discussion and appropriate conclusion was provided. 

Author Response

Dear reviewer,

thank you very much for your specifications. Below are the clarifications to your points:

*Nurses from different educational levels did not seem, eventually, to affect our study results.

*The main issue from the Tercap tool was the question ''During your professional career had ever occured any error in your workplace?'' which was associated to our basic tool (GHQ scale). So, we described the association results of the Tercap specific variables from lines 115-122. Additionally, the person who involved in an adverse event, the complication in health condition, the department with most frequent errors, the criminal intention and the type of the most frequent error (drug error) are some descriptive variables from the Tercap tool. This tool is quite extensive and mostly descriptive and it would be of best practice for the reader not to insert all of the percentages and focus on the associations to general health and resilience level that came up for our study.  

*Corrected-364 nurses

*Corrected-total number of participants

Reviewer 2 Report

I ve read with a great interest the paper entitled "Investigation of mental and physical health of nurses associated to errors in clinical practice". The paper is coherent to the scope of the journal aiming to  to investigate the association between nurses’ general health and making errors during clinical practice. 
The introduction is well written describing the general issue of the topic studied in the paper.
However I need some clarifications in the used methodology:
the anonymous questionnaire were performed from November 2019 to November 2020 and the authors highlighted that due worldwide pandemic, "it was necessary to create a 77 different way of questionnaire distribution and collection, so an electronic form of the tool 78 was developed too". How many questionnaire were send in a way and how many in the other way and how many of which one were completed.
With regards to the non greek native speaker (which nationality? are differences among greek and non greek?
In tables 3 and 4 what do the authors mean with (reference)?
Discussion and conclusion are well written.
References in the text they should be correctly formatted [...]
The reference list should be formatted according the authors guidelines of the journal.

Author Response

Dear reviewer,

thank you very much for your specifications! Below are some clarifications to your points:

  • Questionnaires distributed (manually): 238  (192 received complete)
  • Questionnaires distributed (electronically): 202 (172 received with final ''submit'' choice) 

*Since we had only a few non native greek speakers, there was no difference observed in our study results.

*(Reference) removed. Used only for statistical purposes.

*References Checked
